# STR-Bamba: Multimodal Molecular Textual Representation Encoder-Decoder Foundation Model

## Abstract

Most large-scale chemical language models are trained on a single textual molecular representation using self-supervised learning over large unlabeled corpora. These models excel in tasks such as property prediction and molecule generation by learning contextualized representations of input tokens. However, relying solely on one representation may result in the loss of structural or semantic information captured by alternative formats and may limit the model's ability to generalize across diverse molecular encodings. To address this limitation, we incorporate multiple textual molecular representations—including SMILES, SELFIES, molecular formula, IUPAC name, International Chemical Identifier (InChI), serialized polymer graph (SPG), and electrolyte formulations in an unified vocabulary to harness the unique strengths of each format. Here, we introduce a large encoder-decoder chemical foundation model based on the Bamba architecture, a hybrid of Transformers and Mamba-2 layers, designed to support multi-representational inputs. The model is pre-trained in a BERT-style on 588 million samples, resulting in a corpus of approximately 29 billion molecular tokens. These models serve as a foundation for language chemical research in supporting different complex tasks, including molecular properties prediction, classification, and molecular translation. Furthermore, extensive studies of the multimodal molecular latent space indicate cross-representation alignment and reveal how different textual encodings of the same molecule can converge toward a unified semantic representation. This shared space may facilitate deeper insights into molecular structure, enhance generalization, and support a broad range of downstream applications.

## 1 Introduction

The development of large-scale pre-training methodologies for chemical language models (LMs) constitutes a significant advancement in the field of cheminformatics Sadybekov & Katritch (2023). These methodologies demonstrate a notable efficacy in addressing complex molecular tasks, including the prediction of properties and the generation of molecules Ross et al. (2022b); Soares et al. (2023b). The effectiveness of these models is mainly due to their ability to acquire contextualized representations of input tokens through self-supervised learning on extensive unlabeled corpora Bommasani et al. (2021).

With the advance of the Transformers architecture, several chemical models have been proposed to leverage attention as its core module Pesciullesi et al. (2020); Chithrananda et al. (2020); Janakarajan et al. (2023); Soares et al. (2023a; 2025b). The effectiveness of self-attention is attributed to its ability to route information densely within a context window Vaswani et al. (2017), allowing it to model complex data Tay et al. (2022). However, this property presents essential limitations, such as the inability to model anything outside of a finite window and the quadratic scaling with respect to the window length Lin et al. (2022). A considerable amount of research has emerged on more efficient variants of attention to overcome these drawbacks Kotei & Thirunavukarasu (2023).

In particular, structured state-space sequence models (SSMs) have been introduced as a promising class of architectures to support much longer context lengths for sequence modeling Gu et al. (2021). These models can be interpreted as a combination of recurrent neural networks (RNNs)

and convolutional neural networks (CNNs) Smith et al. (2022). The Mamba model is a simplified end-to-end SSM-based neural network architecture without attention or even MLP blocks Gu & Dao (2023). In this context, recent approaches have demonstrated the efficiency and capability of SSMs in learning a chemical language better than or comparable to Transformer-based models Soares et al. (2025a). To address the limitations and harness the advantages of the self-attention and state-space modules, recent works have proposed a hybrid architecture of Transformer and Mamba-2 layers for general large language models (LLMs) Lieber et al. (2024); Ren et al. (2025); Zhu et al. (2025).

Most of the proposed chemical foundation models rely solely on a single representation or require the adaptability of a new notation for model compatibility. Common molecular representations the models are trained on include SMILES, SELFIES, and PSMILES. However, the use of specific notation may limit the ability of the model to generalize across diverse molecular encodings. Furthermore, diverse molecular notations may complement important molecular information that a specific one does not contain. For PSMILES we used the serialized polymer graph (SPG) representation since it could accommodate both a broad array of polymer architectures and be easily interoperable with existing literature datasets—simplifying assembly of pre-training and benchmarking datasets Soares et al..

In this study, we present a novel hybrid architecture of a large general string-based molecular foundation model of the Transformer and Mamba-2 layers, denoted STR-Bamba$_{426M}$. Our STR-Bamba$_{426M}$ encoder-decoder foundation model leverages multiple molecular string textual representations in a single vocabulary using an efficient attention and SSM-based model. Our main contributions are:

- We pre-train a large-scale encoder-decoder foundation model for molecules, denoted STR-Bamba$_{426M}$, on more than 117 million small molecules from PubChem Kim et al. (2023), 2 million synthetic and real polymers from the literature Soares et al., and 258 electrolyte formulations Sharma et al. (2023), resulting in 119 million unique molecules. With the multimodal setting, the total training data is composed of 588 million samples, which is equivalent to 29 billion molecular tokens.

- A special custom tokenizer for enconding the different molecular representations individually. We built a custom tokenizer to handle each modality properly in a single unified vocabulary of molecular textual representations.

- Our STR-Bamba$_{426M}$ foundation model is an inference-efficiency hybrid Transformer and Mamba-2 base model of 426 million parameters. The design of the model architecture allows for the use of longer context lengths, opening space to leverage multiple molecular representations into a single input. All used code and checkpoints for these models are in progress to be fully open-sourced.

- We perform extensive experimentation on several classification and regression property prediction tasks from 29 benchmark datasets, covering a wide range of tasks for small molecules, polymer molecules, and electrolyte formulations. We also study the quality of the latent space created by the STR-Bamba$_{426M}$ model to represent the multimodal setting of molecular representations. Furthermore, an evaluation of the capabilities of the encoder-decoder configuration of the proposed architecture is conducted by translating different molecular formats of the same molecule.

## 2 OVERVIEW OF THE PROPOSED APPROACH

The following detail the proposed approach of the STR-Bamba architecture to leverage the multimodal molecular textual representations setting in a unified vocabulary and model.

**Tokenization:** A custom tokenizer was carefully built to encode the seven different molecular representations supported by STR-Bamba appropriately. Specifically, we employed the Byte-Pair Encoding (BPE) tokenization for the main encoding process, and a pre-tokenizer step is performed to handle each modality individually. To identify each modality, we considered the special token representations, i.e., *<smiles>* for SMILES, *<selfies>* for SELFIES, *<iupac>* for IUPAC name, *<inchi>* for InChI, *<formula>* for molecular formula, *<polymer_spg>* for SPG, and *<formulation_start>* for electrolyte formulations.

For SMILES, SELFIES and SPG we used the regular expression from Schwaller et al. (2018) to split in an atom-wise approach since it has been extensively used for molecular models Schwaller et al.

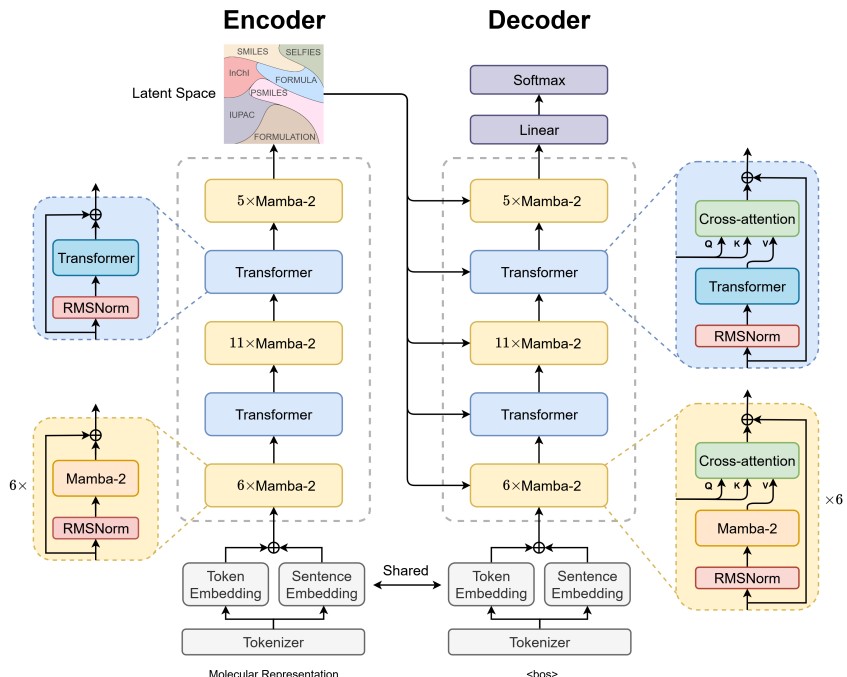

Figure 1: This figure illustrates the general architecture of the base STR-Bamba$_{426M}$ model.

(2019); Irwin et al. (2022); Ross et al. (2022a); Soares et al. (2025b). For the molecular formula and InChI notations, we first extract the numbers associated with the atoms. Finally, formulations use the same atom-wise tokenization with the addition of recognizing formulation compositions. Although the IUPAC name is not preprocessed beforehand, the BPE appropriately encodes the high-frequency pieces of the name. We trained the tokenizer with 5% of the total training data, resulting in approximately 28 million samples for all molecular modalities. The size of the constructed vocabulary yields 5000 tokens, with 13 special tokens and 4987 molecular tokens.

**Pre-training Data:** A combination of small and polymer molecules, and electrolyte formulation data was used to compose our training data. The data on small molecules were extracted from the PubChem database, resulting in a total of 118 million molecules. However, since it does not contain the SELFIES representations, we generated the SELFIES format from the SMILES notation. Hence, the remaining data consist of 117 million molecules with a minimum loss of the total extracted data.

The collection of polymer data for pre-training is a composition of synthetics with experimental datasets, forming a dataset of approximately 2 million polymer structures. The use of generated polymer data is known to contain non-viable polymer structures. Hence, the pre-training data were carefully pooled from a selection of open literature sources, Aldeghi & Coley (2022); Yang et al. (2022); Long et al. (2024); Reis et al. (2021); Tiwari et al. (2024); Giro et al. (2023); Tao et al. (2021); Lo et al. (2023); Hatakeyama-Sato et al. (2023); Huang et al. (2023); Bradford et al. (2023); Xu et al. (2023), avoiding whenever possible, datasets containing potentially problematic structure. We also enriched the training data with an additional 258 electrolyte formulations. Therefore, the total dataset size for pre-training the STR-Bamba$_{426M}$ model comprises approximately 588 million samples. From this total, each of the five molecular modalities from PubChem is used, resulting in 585M samples added with the polymer and formulation data.

**Model Architecture:** STR-Bamba$_{426M}$ is an encoder-decoder model built on the Bamba architecture [1], an inference-efficient hybrid approach of Transformer and Mamba-2. This architecture leverages the strengths of both attention and state-space mechanisms. We slightly modified the Bamba architecture to handle multiple molecular representations and exploit the multimodal setting

---

[1] https://huggingface.co/blog/bamba

into a single unified vocabulary and model, which are shown in Fig. 1. The model configuration for the base architecture, Mamba-2, and Transformer used in our implementation is shown in Table 1.

Table 1: STR-Bamba$_{426M}$ base architecture specificity.

| Hidden size | Layers | lr start | Vocab size | Dataset size | # tokens | # Encoder | # Decoder | Total params |
|---|---|---|---|---|---|---|---|---|
| 1024 | 24 | 3e-5 | 5000 | 119M | 29B | 163M | 263M | 426M |

| | d state | d conv | head dim | expand factor | dt min | dt max | dt init floor |
|---|---|---|---|---|---|---|---|
| | 128 | 4 | 64 | 2 | 0.001 | 0.1 | 1e-4 |

| attn layer index | head dim | num heads | num heads kv | out proj bias | qkv proj bias | rotary emb size |
|---|---|---|---|---|---|---|
| 6, 18 | 64 | 16 | 8 | false | false | 64 |

To align different representations of the same molecule, we performed a modification of the embedding layer similar to the BERT model. To achieve this, we trained the encoder with an aggregation of token and sentence embeddings. The token embedding learns to encode each molecular token properly, and the sentence embedding learns to align one molecular input concatenated to another or a series of representations separated by the *<sep>* special token. In addition, there is no need to add positional encodings after the embedding layer, since Mamba operates in a recurrent way. Finally, the embeddings are shared between the encoder and the decoder to take advantage of the embeddings learned from the encoder.

Following the Bamba model specificity, we placed two attention layers at the beginning and end of the total of 24 layers. Specifically, one attention is followed by 6 layers of Mamba-2 and the other by 18 layers. Additionally, Grouped Query Attention (GQA) and Rotary Position Embeddings (RoPE) are employed in the attention mechanisms to improve training and inference efficiency without losing performance. The use of RoPE embeddings is also exploited to further optimize the relative encoding through position-dependent rotations $R_m$ of the query and keys at position $m$. These rotations can be implemented as pointwise multiplications and do not significantly increase computational complexity as shown in Eq. (1).

$$Attention_m(Q, K, V) = \frac{\sum_{n=1}^{N} \langle \varphi(R_m q_m), \varphi(R_n k_n) \rangle v_n}{\sum_{n=1}^{N} \langle \varphi(q_m), \varphi(k_n) \rangle} \tag{1}$$

where $Q$,$K$,$V$ are the query, key and value, respectively, and $\varphi$ is a random feature map.

Since the base Bamba architecture is a decoder-only model, we also added a cross-attention layer after each Mamba-2 and Transformer layers of the decoder to construct an effective encoder-decoder architecture. The addition of the cross-attention layer is incorporated to generate valid molecular notations conditioned with the embeddings from the encoder. In the implementation, we used the same cross-attention mechanism as in the BART model, receiving the contextual queries and keys from the encoder's embeddings Lewis et al. (2019).

For the state-space layers, we specifically used the Mamba-2 architecture Dao & Gu (2024). The Mamba-2 is an improvement of the original Mamba work by simultaneously allowing much larger state dimensions and reducing the training. The Mamba models originate from a continuous-time system that maps an input function or sequence $x(t) \in \mathbb{R}^M$ to an output response signal $y(t) \in \mathbb{R}^O$ through an implicit latent state $h(t) \in \mathbb{R}^N$ which can be mathematically formulated using the following ordinary differential equations.

$$\begin{aligned} h'(t) &= Ah(t) + Bx(t), \\ y(t) &= Ch(t) + Dx(t) \end{aligned} \tag{2}$$

where $A \in \mathbb{R}^{N \times N}$ and $C \in \mathbb{R}^{O \times N}$ control how the current state evolves over time and translates to the output, $B \in \mathbb{R}^{N \times M}$ and $D \in \mathbb{R}^{O \times M}$ depict how the input influences the state and the output, respectively.

The tokens extracted from the molecular representations through the hybrid Transformer and SSM encoder are embedded in a 1024-dimensional space. Furthermore, each encoder-decoder layer

is designed to process the molecular token embeddings, represented as $\mathbf{x} \in \mathbb{R}^{T \times L}$, where $T$ denotes the input tokens, and $L$ represents the dimension of the embedding space. The length of $T$ has no theoretical limit except for hardware limitations, which opens space to leverage multiple representations in a single input string text.

**Pre-training strategies:** The STR-Bamba$_{426M}$ model was pre-trained in a two-stage strategy. We first train the encoder part to construct a strong embedding space representation for all molecular modalities. Finally, the decoder is trained using the contextual representation of the encoder to correctly predict the next token generation. We used 396 and 8 NVIDIA A100 (40GB) GPUs to train phase 1 and phase 2, respectively. Each phase is described in the following:

- Phase 1 consists of training only the encoder to better learn to encode and align different molecular formats. We employ a similar strategy defined in Devlin et al. (2019) using token and sentence embedding. The token embedding processes the molecular tokens, while the sentence embedding handles a boolean value for each token to assess whether a molecular format $B$ is equivalent to format $A$ for depicting the same molecule. We also used the masked language model from Devlin et al. (2019) to train in a self-supervised way. Thus, the objective of encoder training is to learn to correctly classify masked tokens and to determine if the different molecular formats $A$ and $B$ are the same molecule or not.
- Phase 2 consists of training only the decoder by generating a valid molecular representation given the contextual embeddings of the encoder. To achieve this, we build a batch consisting of reconstructing the input molecule format with the addition of two representations randomly selected of the same molecule. Representations that do not have more than one format are trained to only reconstruct the input text.

## 3 EXPERIMENTS

To evaluate the capability of the STR-Bamba$_{426M}$ model in harnessing all molecular modalities, we performed a series of experiments for all types of molecular notation. An analysis of the latent space is performed to evaluate the effectiveness of the encoder in representing each molecular modality appropriately. For this, we plotted the latent space with t-SNE using 2000 random samples of each modality, except for the electrolyte formulation, we used all 258 samples. A K-means algorithm was used to cluster the semantic regions by varying the number of clusters from 2 to 10. The goal is to evaluate whether a clustering algorithm is capable of recognizing the seven different representations the model supports. For this experiment, we evaluated it using the following clustering metrics: Davies-Bouldin Index, Adjusted Rand Index, V-Measure, and Fowlkes-Mallows Score.

We also evaluated the performance of the STR-Bamba$_{426M}$ model on a wide range of property prediction tasks on 29 datasets, giving a total of 99 tasks. To assess the performance of the model on the data trained in the PubChem database in downstream tasks, we used the MoleculeNet benchmark. To take advantage of the multimodal setting of molecular representations, we assessed each individual and a combination of modalities in the same input text combining the molecular information strengths of each. Thus, we determined all possible combinations between formats and performed an optimization with the Tree-Structured Parzen Estimator (TPE) algorithm using the validation set to find the top-3 combinations for each task.

To evaluate the performance of property predictions for polymer structures, we employed a variety of benchmark datasets sourced from the existing literature. We also used the dataset benchmarks from Sharma et al. (2023) to assess the electrolyte formulation in property prediction tasks. All experiments were carried out with five different seeds to ensure the statistical relevance of the results. In addition, to ensure an unbiased assessment, we maintained consistency with the original benchmark by adopting identical train/validation/test splits for all tasks. Detailed specification for each benchmark dataset and evaluation metrics used is provided in the Supplementary Materials.

Finally, the encoder-decoder architecture of STR-Bamba enables a wide range of tasks. Therefore, we also assessed the ability of the decoder to translate a molecular representation into another in the same molecule. For this, we evaluated the model on 3007 random molecules from the training data to generate valid and structure similarity SMILES and SELFIES using the *RDKit*[2] library and tanimoto

---
[2]https://www.rdkit.org

similarity, respectively. Additionally, the generation of the IUPAC name text and the molecular formula was assessed using the BLEU-1, BLEU-2 and Jaccard similarity.

# 4 RESULTS AND DISCUSSION

In this section, we provide a wide range of experimental results for the STR-Bamba$_{426M}$ architecture, accompanied by a discussion. The experiments consist of: i) A latent space analysis of multiple molecular representation; ii) Performance assessment on various property prediction tasks; iii) Translation of different representations of the same molecule.

## 4.1 LATENT SPACE STUDY

To evaluate the effectiveness of the encoder in learning the seven molecular representations, the K-means clustering algorithm is used to delimit the different regions in the latent space. Figure 2a shows the projection of the t-SNE of the encoder embeddings for each molecular format and Fig. 2b shows seven clusters determined by the K-means algorithm.

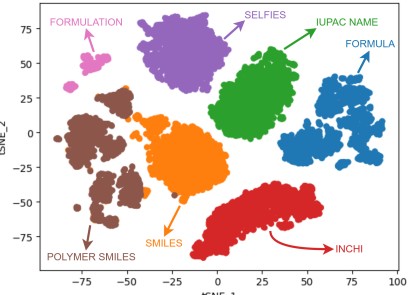 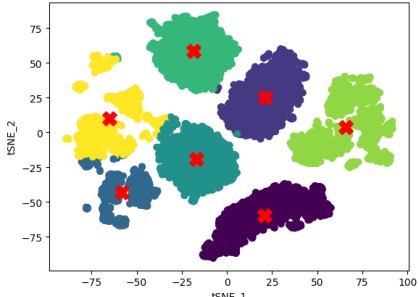

(a) Latent space of the 7 different molecular modalities.   (b) Identified clusters using the K-means algorithm.

Figure 2: Latent space analysis of multiple molecular representation.

Table 2: Performance of K-means latent space clustering.

| Number of clusters ($n$) | $n$=2 | $n$=3 | $n$=4 | $n$=5 | $n$=6 | $n$=7 | $n$=8 | $n$=9 | $n$=10 |
|---|---|---|---|---|---|---|---|---|---|
| Davies-Bouldin Index ↓ | 1.18 | 0.87 | 0.73 | 0.75 | 0.81 | 0.70 | **0.69** | 0.71 | 0.71 |
| Adjusted Rand Index ↑ | 0.26 | 0.41 | 0.61 | 0.74 | 0.84 | **0.91** | 0.84 | 0.79 | 0.79 |
| V-Measure ↑ | 0.43 | 0.57 | 0.74 | 0.82 | 0.88 | **0.92** | 0.88 | 0.87 | 0.87 |
| Fowlkes-Mallows Score ↑ | 0.51 | 0.58 | 0.71 | 0.80 | 0.86 | **0.92** | 0.87 | 0.83 | 0.83 |

In Table 2 the number of clusters is varied between distinct clustering metrics to systematic determine the best number of regions. The Davies-Boulding Index shows that eight clusters yield the most perfect match followed by seven, nine, and ten regions, respectively. Similarly, the Adjusted Rand Index, V-Measure, and Fowlkes-Mallows Score exhibit the best delimitation by seven clusters. This shows that by clustering the latent space achieves the same number of molecular representations as STR-Bamba$_{426M}$ supports.

It is noteworthy that the metrics employed suggest that eight clusters also have a good clustering determination. This can be seen as the PSMILES represented by the SPG notation and SMILES have an intersection, which is seen to be natural since both have similar textual appearances with slightly different notation.

## 4.2 COMPARISON WITH SOTA ON PROPERTY PREDICTION BENCHMARKING TASKS

**MoleculeNet benchmark:** An assessment of the learned multimodal latent space for property prediction is conducted for small molecules using the MoleculeNet benchmark across various tasks. Tables 9 and 10 show the performance comparison between the STR-Bamba$_{426M}$ model and the latest models in the literature for classification and regression, respectively. For our model, we individually

tested each molecular representation and the top-3 combination of formats. Thus, these tables show the best performing results. Detailed results for all individual and combined notations tested, and a full comparison with SOTA models can be found in the Supplementary Materials.

Table 3: Methods and Performance for the classification tasks of MoleculeNet benchmark datasets. **Blue** and **Orange** indicates best and second-best performing model, respectively.

| Method | Dataset | | | | | |
|---|---|---|---|---|---|---|
| | BBBP | ClinTox | HIV | BACE | SIDER | Tox21 |
| SOTA | 92.81±0.27 | **90.8±8.1** | 83.14±0.34 | **89.0±0.3** | **68.0±1.1** | **83.84±0.2** |
| STR-Bamba$_{426M}$ (Pre-trained) | **93.85±0.5** | 93.32±0.9 | **83.14±0.62** | 85.6±0.67 | 66.42±0.38 | 81.57±0.37 |
| STR-Bamba$_{426M}$ (Fine-tuned) | **94.30±0.61** | **96.44±0.07** | **84.77±0.46** | **89.84±0.23** | **69.41±0.62** | **85.02±0.65** |

Table 4: Methods and Performance for the regression tasks of MoleculeNet benchmark datasets. **Blue** and **Orange** indicates best and second-best performing model, respectively.

| Method | Dataset | | | | |
|---|---|---|---|---|---|
| | QM9 | QM8 | ESOL | FreeSolv | Lipophilicity |
| SOTA | **1.3246±0.0157** | **0.0095±0.0001** | 0.6112±0.0096 | **1.2233±0.0029** | **0.532±0.013** |
| STR-Bamba$_{426M}$ (Pre-trained) | 6.8618±0.0538 | 0.0176±0.0002 | **0.6199±0.0457** | 1.3989±0.0837 | 0.6825±0.0061 |
| STR-Bamba$_{426M}$ (Fine-tuned) | **1.5574±0.0156** | 0.0104±0.0001 | **0.5585±0.0201** | **0.9426±0.0412** | 0.5741±0.0073 |

For classification tasks, the STR-Bamba$_{426M}$ model outperformed five of the six downstream tasks compared to the SOTA models. The ClinTox dataset was the only task surpassed by another model, which is the MetaGIN architecture, a graph-based model. Although the MetaGIN model achieved the best performance in the ClinTox task, our model reached the second-best performance. Furthermore, the variation in MetaGIN results is considerable high with a ROC-AUC average of 90.8%. In contrast, the STR-Bamba model achieved an ROC-AUC average of 96.44% with a very slight variation between different seeds. This can demonstrate some instability of the MetaGIN model in this task, which can occasionally outperform STR-Bamba on the ClinTox task.

Although the STR-Bamba$_{426M}$ model outperformed two of five regression tasks, the model obtained very close results compared to the SOTA models. Our model achieved outstanding performance in the ESOL and FreeSolv tasks. Additionally, it achieved the second best performing model for the ESOL task with the pre-trained model and QM9 dataset with the fine-tuned model.

These results demonstrate the ability of the hybrid approach to perform better or have performance comparable to Transformer-based or SSM-based only models by leveraging multiple formats. Finally, in nine out of 11 downstream tasks evaluated on the MoleculeNet benchmark, the combination of molecular representations obtained the best results with the STR-Bamba$_{426M}$ model. This demonstrates the importance of taking advantage of the strengths of each modality in a unified model.

**Polymer benchmarks:** The STR-Bamba$_{426M}$ model was also evaluated in a wide range of polymer property prediction tasks from the literature. Thus, Figure 3 shows the results in 17 downstream tasks in which the normalized error is considered to assess the model compared to the SOTA models. Similarly, we conducted more 9 polymer property prediction tasks in which the $R^2$ metric was used, resulting in a total of 26 downstream tasks for polymer structures. The results obtained from Fig. 3 and the results of the 9 mentioned polymer prediction tasks are detailed in the Supplementary Materials.

In all 26 polymer tasks, the STR-Bamba$_{426M}$ model outperformed or reached near-state-of-the-art results in 17 downstream tasks. In Fig. 3 with the tasks in which the error was used to evaluate the models, the green area on the left shows that the model was equal or better than the SOTA models in 10 of 17 property prediction tasks.

In the nine remaining polymer tasks, especially for polymer membrane tasks, the STR-Bamba$_{426M}$ model notably outperformed the models documented in the literature. In tasks $T_{d\frac{1}{2}}$ and $\log(P_{CO_2})$, the pre-trained model achieved a better performance than SOTA models with an additional improvement from fine-tuned models.

Finally, for the gas permeability of polymer tasks (CalTech), the STR-Bamba$_{426M}$ model was capable of outperforming or achieving SOTA results in 4 out of 6 tasks and achieved the second best performance for the remaining two tasks. Although STR-Bamba did not surpass the DNN

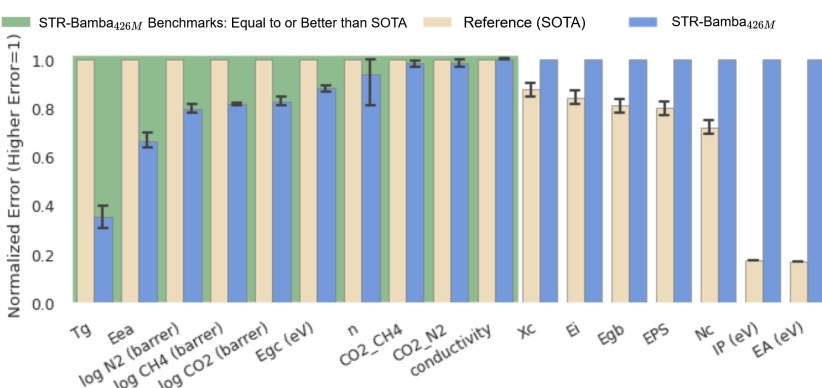

Figure 3: Comparison of the STR-Bamba$_{426M}$ model with state-of-the-art models across various polymer property predictions. The results show that STR-Bamba$_{426M}$ outperforms SOTA models in **10 out of 17 properties**. The errors are normalized such that a value of 1 represents the maximum error observed in the comparison.

ensemble(MFFs) model on the $O_2$ task with a $R^2$ of 0.92, it reached a very close performance with a $R^2$ average of 0.91. Similarly, for the $CO_2$ task, our model achieved an average $R^2$ of 0.90, while the SPG-TED$_{289M}$ obtained a $R^2$ of 0.91.

This may illustrate the richness capability of the latent space of STR-Bamba in learning multiple molecular formats. Therefore, the shared common space of diverse chemical representations may enhance the molecular prediction, in which the SMILES share some properties with SPG as seen in the latent space study.

**Electrolyte formulation benchmarks:** We also evaluated our model on electrolyte formulation tasks for property prediction. In particular, we used two datasets to predict the LCE of formulations and the specific battery capacity. The best results for the STR-Bamba$_{426M}$ model compared to the SOTA models are shown in Table 5. Detailed results for each individual format are provided in the Supplementary Materials.

Table 5: Electrolyte formulation prediction performance. RMSE is used as evaluation metric, therefore, in this case lower is better. **Blue** and **Orange** indicates best and second-best performing model, respectively.

| Method | Dataset | |
|---|---|---|
| | Li/Cu Half-Cell (LCE) | Li-I Full-Cell Battery (Capacity) |
| MoLFormer Aldeghi & Coley (2022) | 0.213 | - |
| Multimodal MoLFormer Aldeghi & Coley (2022) | **0.195** | - |
| F-GCN Sharma et al. (2023) | 0.389 | 39.823 |
| F-GCN with HL-EM descriptors Sharma et al. (2023) | - | **20.495** |
| STR-Bamba$_{426M}$ (Pre-trained) | 0.235±0.017 | 33.174±3.007 |
| STR-Bamba$_{426M}$ (Fine-tuned) | **0.214±0.031** | **32.496±7.066** |

From the two tasks, the STR-Bamba$_{426M}$ model was able to surpass the SOTA models in the LCE task and reach the second-best model for the specific battery capacity task. The significant variation in the results may be due to the limited data size of no more than 150 samples for each of both datasets, which reinforces the need for statistical validity of the results. In the LCE task, the fine-tuned model had slightly higher RMSE average compared to the Multimodal MoLFormer and MoLFormer models. However, due to the high standard deviation, our model was able to outperform with the SELFIES notation the Multimodal MoLFormer, which does not provide any variation in the results.

For the specific battery capacity task, the STR-Bamba$_{426M}$ model outperformed the F-GCN model but achieved the second best performance with the InChI notation of an average RMSE of 32.496 comparing the variant of F-GCN with the HOMO-LUMO (HL) and electric moment (EM) molecular descriptors of an RMSE of 20.495. Furthermore, the results demonstrate the capability of the diverse

multimodal latent space of our model, since the STR-Bamba$_{426M}$ model was pre-trained in a limited sample of only 258 electrolyte formulations.

### 4.3 REPRESENTATION TRANSLATION

The encoder-decoder architecture setting of STR-Bamba gives flexibility to a range of downstream tasks. Hence, we also evaluated the performance of the STR-Bamba$_{426M}$ model in translating between different representations of the same molecule. The results of translating a representation to SMILES and SELFIES are shown in Fig. 4a, while the translation of a representation to IUPAC name and molecular formula are shown in Fig. 4b.

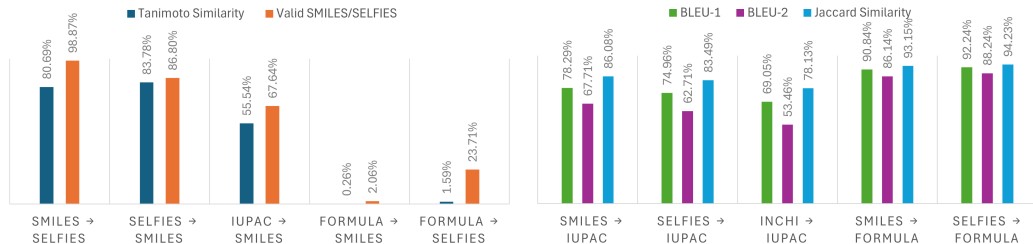

(a) Translation to SMILES and SELFIES.  (b) Translation to IUPAC and molecular formula.

Figure 4: Translation of different molecular representations.

As the decoder training was conducted without using the entire training data, all results show with a few-shot learning approach. For translating from SMILES to SELFIES and vice versa, the representations achieved the best structural and validity generation, which can be explained that SELFIES is a notation derived from SMILES notation. Similarly, the IUPAC name format may generate a similar SMILES notation but with less structure and valid molecules guarantees. In addition, the model was tasked with generating SMILES and SELFIES from the molecular formula notation. This can be a challenging task due to multiple valid molecules with different structure properties that can be composed from the molecular formula. However, it is noteworthy that the SELFIES notation generated a considerable number of valid molecules compared to SMILES since this representation was developed to be a robust molecular representation.

The task of generating the molecular formula from SMILES and SELFIES achieved results very close to the original formula. In particular, the translation from the SELFIES notation was slightly better compared to SMILES representation. However, generating the IUPAC name was a more difficult task. The SMILES notation achieved the best results from this task, whereas the InChI achieved the lowest. In general, the performance achieved in the representation translation task shows the potential ability of the STR-Bamba$_{426M}$ model to generate similar and valid molecular representations in the same molecule. The proposed architecture of multiple molecular formats in a unified latent space helps the model align the different modalities in generation tasks.

## 5 CONCLUSION

This paper introduces the STR-Bamba$_{426M}$ model, a multimodal textual molecular representation foundation model of a hybrid Transformer and Mamba-2 architecture capable of encoding multiple molecular notations in a single model. A custom tokenizer was developed to allow the encoding of each modality appropriately for the model. Additionally, the STR-Bamba architecture allows for the aggregation of multiple representations in a single text input, as it does not contain any token length limitation, except for hardware limitations.

Extensive experimentation with prediction of the molecule properties of small molecules, polymers, and electrolyte formulations achieved competitive results by leveraging the multimodal setting compared to state-of-the-art models. Furthermore, the latent space analysis demonstrates the model's capability to represent each molecular format. Finally, the encoder-decoder architecture allows multiple tasks, such as translating between representations of the same molecule, showing the potential to walk between modalities.

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

## A SUPPLEMENTARY MATERIALS

### A.1 PROPERTY PREDICTION BENCHMARKS DETAILS

Here, we provide detailed results for all the property prediction experiments conducted in this paper. To ensure the robustness of our claims, we conducted all experiments with five different seeds. For

training models with pre-trained weights, we utilized XGBoost Chen et al. (2015) as the learner and Optuna Akiba et al. (2019) for hyper-parameter optimization. All experiments with pre-trained models were conducted using a single NVIDIA A100 (40G) GPU.

For fine-tuning STR-Bamba$_{426M}$, we used a fully connected network with 2 layers using a single NVIDIA A100 (40G) GPU. Tables 6, 7, and 8 provide a detailed overview of small and polymer molecules, and electrolyte formulation benchmark datasets used in our experiments, respectively.

Table 6: Evaluated small molecular datasets description

| Dataset | Description | # compounds | # tasks | Metric |
|---|---|---|---|---|
| BBBP | Blood brain barrier penetration dataset | 2039 | 1 | ROC-AUC |
| HIV | Ability of small molecules to inhibit HIV replication | 41127 | 1 | ROC-AUC |
| BACE | Binding results for a set of inhibitors for $\beta-$secretase 1 | 1513 | 1 | ROC-AUC |
| Clintox | Clinical trial toxicity of drugs | 1478 | 2 | ROC-AUC |
| SIDER | Drug side effect on different organ classes | 1427 | 27 | ROC-AUC |
| Tox21 | Toxicity measurements on 12 different targets | 7831 | 12 | ROC-AUC |
| QM9 | 12 quantum mechanical calculations | 133885 | 12 | Average MAE |
| QM8 | 12 excited state properties of small molecules | 21786 | 12 | Average MAE |
| ESOL | Water solubility dataset | 1128 | 1 | RMSE |
| FreeSolv | Hydration free energy of small molecules in water | 642 | 1 | RMSE |
| Lipophilicity | Octanol/water distribution coefficient of molecules | 4200 | 1 | RMSE |

Table 7: Evaluated polymer molecular datasets description

| Dataset | Description | Metric | Source |
|---|---|---|---|
| Copolymers (MIT) | DFTB computed electron affinity and ionization potential of copolymers. | RMSE | Aldeghi & Coley (2022) |
| IBM-Membrane | Computed thermal and gas permeability properties of polymers. | $R^2$ | Giro et al. (2023) |
| ACS-AMI-Homopolymer-Tg | Tg of homopolymers | RMSE | Hu et al. (2023) |
| Polymer-Refractive-Index | Polymer refractive index | RMSE | Hatakeyama-Sato et al. (2023) |
| Polymer-Electrolyte-Conductivity (MIT) | Conductivity of polymers and polymer formulations | MAE | Bradford et al. (2023) |
| Polymer-Gas-Permeability (NETL) | Gas permeability and selectivity of polymers | MAE | Tiwari et al. (2024) |
| Polymer-Gas-Permeability (CalTech) | Gas permeability of polymers | $R^2$ | Yang et al. (2022) |
| Polyimide-Tg | Tg of polyimides | MAE | Long et al. (2024) |
| Polymer-Chain-Bandgap-(Egc) | DFT computed polymer chain bandgap | RMSE | Kuenneth et al. (2021) Xu et al. (2023) |
| Polymer-Electron-Affinity-(Eea) | DFT computed electron affinity of polymers | RMSE | Kuenneth et al. (2021) Xu et al. (2023) |
| Polymer-Bulk-Bandgap-(Egb) | DFT computed bulk bandgap of polymers | RMSE | Kuenneth et al. (2021) Xu et al. (2023) |
| Polymer-Ionization-Energy-(Ei) | DFT computed ionization energy of polymers | RMSE | Kuenneth et al. (2021) Xu et al. (2023) |
| Polymer-Dielectric-Constant-(EPS) | DFT computed dielectric constant of polymers | RMSE | Kuenneth et al. (2021) Xu et al. (2023) |
| Polymer-Refractive-Index-(Nc) | DFT computed refractive index of polymers | RMSE | Kuenneth et al. (2021) Xu et al. (2023) |
| Polymer-Crystallization-Tendency-(Xc) | DFT computed crystallization tendency of polymers | RMSE | Kuenneth et al. (2021) Xu et al. (2023) |
| Polymer-Conductivity-(PE-II) | Conductivity of polymers | RMSE | Xu et al. (2023) |

Table 8: Evaluated electrolyte formulation datasets description

| Dataset | Description | Metric | Source |
|---|---|---|---|
| Li/Cu Half-Cell | Logarithmic Coulombic efficiencies (LCE) of a wide range of electrolyte formulations | RMSE | Aldeghi & Coley (2022) |
| Li-I Full-Cell Battery | Specific capacities of Li-I battery coin cells | RMSE | Giro et al. (2023) |

## A.2    DETAILED RESULTS - FULL COMPARISON WITH SOTA MODELS IN MOLECULENET BENCHMARK

We provide the detailed comparison with the SOTA models in the MoleculeNet benchmark. Thus, Table 9 and 10 show the full comparison with the STR-Bamba$_{426M}$ pre-trained and fine-tuned models in the classification and regression tasks, respectively.

Table 9: Methods and Performance for the classification tasks of MoleculeNet benchmark datasets. **Blue** and **Orange** indicates best and second-best performing model, respectively.

| Method | Dataset | | | | | |
|---|---|---|---|---|---|---|
| | BBBP | ClinTox | HIV | BACE | SIDER | Tox21 |
| GraphMVP Liu et al. (2021) | 72.4±1.6 | 79.1±2.8 | 77.0±1.2 | 81.2±0.9 | 63.9±1.2 | 75.9±0.5 |
| GEM Fang et al. (2022) | 72.4±0.4 | 90.1±1.3 | 80.6±0.9 | 85.6±1.1 | 67.2±0.4 | 78.1±0.1 |
| GROVER$_{Large}$ Rong et al. (2020) | 69.5±0.1 | 76.2±3.7 | 68.2±1.1 | 81.0±1.4 | 65.4±0.1 | 73.5±0.1 |
| ChemBerta Chithrananda et al. (2020) | 64.3 | 90.6 | 62.2 | - | - | - |
| ChemBerta2 Ahmad et al. (2022) | 71.94 | 90.7 | - | 85.1 | - | - |
| Galatica 30B Taylor et al. (2022) | 59.6 | 82.2 | 75.9 | 72.7 | 61.3 | 68.5 |
| Galatica 120B Taylor et al. (2022) | 66.1 | 82.6 | 74.5 | 61.7 | 63.2 | 68.9 |
| Uni-Mol Zhou et al. (2023) | 72.9±0.6 | 91.9±1.8 | 80.8±0.3 | 85.7±0.2 | 65.9±1.3 | 79.6±0.5 |
| MolCLR$_{GIN}$ Wang et al. (2022) | 73.6±0.5 | 93.2±1.7 | 80.6±1.1 | 89.0±0.3 | 68.0±1.1 | 79.8±0.7 |
| MolFM Zhou et al. (2023) | 72.9±0.1 | 79.7±1.6 | 78.8±1.1 | 83.9±1.1 | 64.2±0.9 | 77.2±0.7 |
| MoLFormer Chang & Ye (2024) | 73.6±0.8 | 91.2±1.4 | 80.5±1.65 | 86.3±0.6 | 65.5±0.2 | 80.46±0.2 |
| MetaGIN Zhang et al. (2024) | 91.7±1.8 | 90.8±8.1 | - | - | 64.5±2.4 | 83.0±0.1 |
| SMI-TED289M Soares et al. (2025b) | 92.26±0.57 | 94.27±1.83 | 80.51±1.34 | 88.24±0.50 | 66.01±0.88 | 81.85±1.42 |
| SMI-SSED$_{336M}$ Soares et al. (2025a) | 92.81±0.27 | 90.02±0.5 | 83.14±0.34 | 86.12±0.96 | 63.17±0.75 | 83.84±0.2 |
| STR-Bamba$_{426M}$ (Pre-trained) | 93.85±0.5 | 93.32±0.9 | 83.14±0.62 | 85.6±0.67 | 66.42±0.38 | 81.57±0.37 |
| STR-Bamba$_{426M}$ (Fine-tuned) | 94.30±0.61 | 96.44±0.07 | 84.77±0.46 | 89.84±0.23 | 69.41±0.62 | 85.02±0.65 |

Table 10: Methods and Performance for the regression tasks of MoleculeNet benchmark datasets. **Blue** and **Orange** indicates best and second-best performing model, respectively.

| Method | Dataset | | | | |
|---|---|---|---|---|---|
| | QM9 | QM8 | ESOL | FreeSolv | Lipophilicity |
| D-MPNN Yang et al. (2019) | 3.241±0.119 | 0.0143±0.0022 | 0.98±0.26 | 2.18±0.91 | 0.65±0.05 |
| N-Gram Liu et al. (2019) | 2.51±0.19 | 0.032±0.003 | 1.074±0.107 | 2.688±0.085 | 0.812±0.028 |
| PretrainGNN Hu et al. (2019) | - | - | 1.100±0.006 | 2.764±0.002 | 0.739±0.003 |
| GROVER$_{Large}$ Rong et al. (2020) | - | - | 0.895±0.017 | 2.272±0.051 | 0.823±0.010 |
| ChemBERTa-2 Ahmad et al. (2022) | - | - | 0.89 | - | 0.80 |
| SPMM Chang & Ye (2024) | - | - | 0.818±0.008 | 1.907±0.058 | 0.692±0.008 |
| MolCLR$_{GIN}$ Wang et al. (2022) | 2.357±0.118 | 0.0174±0.0013 | 1.11±0.01 | 2.20±0.20 | 0.65±0.08 |
| Hu et al. Hu et al. (2020) | 4.349±0.061 | 0.0191±0.0003 | 1.22±0.02 | 2.83±0.12 | 0.74±0.00 |
| MoLFormer Chang & Ye (2024) | 1.5894±0.0567 | 0.0102 | 0.880±0.028 | 2.342±0.052 | 0.700±0.012 |
| MetaGIN Zhang et al. (2024) | - | - | 0.780±0.061 | 1.397±0.062 | 0.532±0.013 |
| SMI-TED289M Soares et al. (2025b) | 1.3246±0.0157 | 0.0095±0.0001 | 0.6112±0.0096 | 1.2233±0.0029 | 0.5522±0.0194 |
| SMI-SSED$_{336M}$ Soares et al. (2025a) | 2.2175±0.3194 | 0.0104±0.0001 | 0.7222±0.0139 | 1.6374±0.0682 | 0.6048±0.0023 |
| STR-Bamba$_{426M}$ (Pre-trained) | 6.8618±0.0538 | 0.0176±0.0002 | 0.6199±0.0457 | 1.3989±0.0837 | 0.6825±0.0061 |
| STR-Bamba$_{426M}$ (Fine-tuned) | 1.5574±0.0156 | 0.0104±0.0001 | 0.5585±0.0201 | 0.9426±0.0412 | 0.5741±0.0073 |

## A.3 DETAILED RESULTS - INDIVIDUAL AND COMBINED MOLECULAR REPRESENTATIONS IN MOLECULENET BENCHMARK

Here we detail the results in the MoleculeNet benchmark for each molecular representation and the top-3 combination of small molecule notations. The results for each modality using pre-trained and fine-tuned models are shown in Tables 11 and 12 for classification and regression tasks, respectively.

We used TPE optimization by evaluating the validation set to find the top-3 combinations of molecular representations that demonstrate the best performance for each task. The optimization process was repeated with three different seeds for each task to ensure the statistical validity of the results. To obtain a single combination for the repeated optimization steps, we performed an intersection of the top combinations between each optimization execution. Hence, combination 1 shows the top-1 combination, combination 2 show the top-2 and combination 3 show the top-3 combination. To separate each chemical representation for the fused molecular input, we used the *<sep>* special token between them. Furthermore, in the sentence embedding from the encoder embedding layer, the first molecule notation was represented as the molecule *A* and the remaining notations as the molecule or a series of molecules *B*.

Finally, we also provide each molecular combination for the top combinations used in the evaluated tasks in the MoleculeNet benchmark. Tables 13, 14, and 15 show the combinations for top-1, top-2, and top-3, respectively. For molecular combinations, we tried all the possible combinations where the order was considered and individual representations are also included, yielding 325 possible combinations.

Table 11: Individual and combined molecular representations performance for the classification tasks of MoleculeNet benchmark datasets

| Representation | Method | Dataset | | | | | |
| --- | --- | --- | --- | --- | --- | --- | --- |
| | | BBBP | ClinTox | HIV | BACE | SIDER | Tox21 |
| Molecular Formula | Pre-trained | 87.67±0.39 | 90.22±0.50 | 71.86±1.01 | 77.49±0.16 | 63.61±0.73 | 74.67±0.86 |
| Canonical SMILES | Pre-trained | 91.27±0.37 | 91.0±0.71 | 80.98±0.69 | 85.60±0.67 | 66.42±0.38 | 80.33±0.39 |
| IUPAC Name | Pre-trained | 93.85±0.50 | 85.28±1.60 | 83.14±0.62 | 69.19±1.44 | 65.05±0.53 | 80.11±0.90 |
| InChI | Pre-trained | 91.08±0.32 | 85.42±1.50 | 78.17±0.95 | 84.04±0.66 | 63.56±0.65 | 79.07±0.46 |
| SELFIES | Pre-trained | 90.66±0.57 | 93.32±0.90 | 79.91±0.97 | 85.18±0.75 | 65.83±0.38 | 79.85±0.72 |
| Combination 1 | Pre-trained | 93.85±0.50 | 90.69±1.0 | 81.30±1.37 | 85.35±0.58 | 66.42±0.38 | 81.57±0.37 |
| Combination 2 | Pre-trained | 90.73±0.46 | 91.79±0.96 | 81.47±0.76 | 85.60±0.67 | 65.60±0.44 | 81.23±0.70 |
| Combination 3 | Pre-trained | 91.25±1.02 | 91.24±0.85 | 82.78±0.86 | 83.82±1.40 | 65.16±0.31 | 81.21±0.47 |
| Molecular Formula | Fine-tuned | 87.74±0.28 | 87.85±0.39 | 72.75±0.59 | 75.99±1.37 | 63.05±1.39 | 80.09±0.39 |
| Canonical SMILES | Fine-tuned | 92.45±0.78 | 94.77±0.73 | 80.88±1.42 | 86.94±0.95 | 67.03±0.76 | 85.07±0.30 |
| IUPAC Name | Fine-tuned | 92.15±0.80 | 91.11±0.80 | 79.46±1.58 | 69.91±1.63 | 67.49±0.91 | 83.41±0.46 |
| InChI | Fine-tuned | 91.31±0.55 | 90.30±1.20 | 77.29±0.96 | 81.91±1.66 | 63.47±0.74 | 82.45±0.19 |
| SELFIES | Fine-tuned | 91.85±0.74 | 96.06±0.31 | 81.91±0.37 | 87.15±0.31 | 66.41±1.06 | 84.39±0.33 |
| Combination 1 | Fine-tuned | 92.15±0.80 | 94.81±1.29 | 84.44±0.31 | 88.15±0.62 | 67.03±0.76 | 85.14±0.51 |
| Combination 2 | Fine-tuned | 92.93±0.48 | 96.44±0.07 | 84.77±0.46 | 86.94±0.95 | 69.41±0.62 | 85.16±0.36 |
| Combination 3 | Fine-tuned | 94.30±0.61 | 95.32±0.93 | 84.46±0.65 | 89.84±0.23 | 68.19±0.55 | 85.02±0.65 |

Table 12: Individual and combined molecular representations performance for the regression tasks of MoleculeNet benchmark datasets

| Representation | Method | Dataset | | | | |
| --- | --- | --- | --- | --- | --- | --- |
| | | QM9 | QM8 | ESOL | FreeSolv | Lipophilicity |
| Molecular Formula | Pre-trained | 13.3875±0.0037 | 0.0297±0.0001 | 0.6694±0.0152 | 1.7005±0.0556 | 0.8124±0.0082 |
| Canonical SMILES | Pre-trained | 6.8618±0.0538 | 0.0176±0.0002 | 0.6439±0.0219 | 1.5269±0.0850 | 0.6888±0.0066 |
| IUPAC Name | Pre-trained | 20.7505±0.0209 | 0.0245±0.0001 | 0.8039±0.0243 | 1.7784±0.0472 | 0.7071±0.0031 |
| InChI | Pre-trained | 6.8553±0.0388 | 0.0210±0.0003 | 0.7048±0.0127 | 1.7284±0.0831 | 0.7289±0.0330 |
| SELFIES | Pre-trained | 6.6983±0.0257 | 0.0178±0.0002 | 0.6679±0.0179 | 1.6113±0.0916 | 0.7096±0.0076 |
| Combination 1 | Pre-trained | 6.5891±0.0355 | 0.0194±0.0008 | 0.6199±0.0457 | 1.4586±0.070 | 0.6950±0.0087 |
| Combination 2 | Pre-trained | 6.8737±0.0260 | 0.0220±0.0020 | 0.6302±0.0103 | 1.5482±0.0721 | 0.6825±0.0061 |
| Combination 3 | Pre-trained | 6.8363±0.0406 | 0.0232±0.0024 | 0.6346±0.0261 | 1.3989±0.0837 | 0.6867±0.0099 |
| Molecular Formula | Fine-tuned | 12.9819±0.0092 | 0.0257±0.00003 | 0.7719±0.0348 | 1.7220±0.1481 | 0.8787±0.0239 |
| Canonical SMILES | Fine-tuned | 1.5574±0.0156 | 0.0107±0.0001 | 0.6073±0.0190 | 1.0909±0.0325 | 0.5741±0.0073 |
| IUPAC Name | Fine-tuned | 18.347±0.0110 | 0.0193±0.0001 | 0.9553±0.0207 | 1.6983±0.0366 | 0.6392±0.0096 |
| InChI | Fine-tuned | 2.3886±0.0619 | 0.0161±0.0001 | 0.7262±0.0256 | 1.2892±0.0216 | 0.6988±0.0115 |
| SELFIES | Fine-tuned | 1.5950±0.0302 | 0.0111±0.0001 | 0.6346±0.0317 | 1.2546±0.0678 | 0.6063±0.0104 |
| Combination 1 | Fine-tuned | 1.5732±0.0338 | 0.0105±0.0001 | 0.5683±0.0168 | 0.9426±0.0412 | 0.5857±0.0032 |
| Combination 2 | Fine-tuned | 1.7374±0.0262 | 0.0104±0.0001 | 0.5514±0.0087 | 1.1049±0.1087 | 0.5914±0.0021 |
| Combination 3 | Fine-tuned | 1.6183±0.0267 | 0.0104±0.0001 | 0.5585±0.0201 | 1.1105±0.0587 | 0.5814±0.0073 |

A.4    DETAILED RESULTS - POLYMER PROPERTY PREDICTION TASKS

Here we provide the detailed results of the nine polymer prediction tasks in which the $R^2$ metric was used and the results for the remaining 17 polymer property prediction tasks, resulting in a total of 26 polymer downstream tasks.

Specifically, Tables 16 and 17 show the results for the polymer membranes and the gas permeability of polymers (CalTech) datasets, respectively. The polymer membrane prediction dataset contains three different tasks. Similarly, the gas permeability of polymers (CalTech) dataset contains six different tasks.

The latter of 17 polymer property prediction tasks comprises a total of six different datasets. Hence, Tables 18, 19, 20, 21, 22, and 23 show the comparison between STR-Bamba$_{426M}$ model with SOTA models for polymer ionic conductivity, gas permeability (NETL), polymer refractive index, polymer multitask prediction, copolymer electron affinity and ionization potential, and glass-transition temperature datasets, respectively.

A.5    DETAILED RESULTS - INDIVIDUAL MOLECULAR REPRESENTATIONS IN ELECTROLYTE
         FORMULATION TASKS

Finally, we detail the results for each individual molecular representation for electrolyte formulation tasks, which are shown in Table 24. Each result was tested on five different seeds to determine the robustness and statistical validity of our experiments. For the electrolyte formulation strings, we placed the special tokens *<formulation_start>* and *<formulation_end>* at the beginning and end of

Table 13: Top-1 molecular combinations for the MoleculeNet benchmark datasets.

| Dataset | Task | Molecular combination |
|---|---|---|
| BBBP | p_np | IUPAC_NAME |
| ClinTox | all | INCHI + MOLECULAR_FORMULA + IUPAC_NAME + SELFIES + CANONICAL_SMILES |
| HIV | HIV_active | CANONICAL_SMILES + SELFIES + MOLECULAR_FORMULA + INCHI + IUPAC_NAME |
| BACE | Class | IUPAC_NAME + MOLECULAR_FORMULA + SELFIES + INCHI + CANONICAL_SMILES |
| SIDER | all | CANONICAL_SMILES |
| Tox21 | all | IUPAC_NAME + MOLECULAR_FORMULA + INCHI + CANONICAL_SMILES + SELFIES |
| QM9 | alpha | IUPAC_NAME + SELFIES + MOLECULAR_FORMULA + INCHI |
| QM9 | cv | SELFIES + MOLECULAR_FORMULA + INCHI |
| QM9 | g298 | SELFIES + MOLECULAR_FORMULA + INCHI + CANONICAL_SMILES |
| QM9 | gap | CANONICAL_SMILES |
| QM9 | h298 | SELFIES + MOLECULAR_FORMULA + INCHI + IUPAC_NAME + CANONICAL_SMILES |
| QM9 | homo | CANONICAL_SMILES |
| QM9 | lumo | CANONICAL_SMILES |
| QM9 | mu | CANONICAL_SMILES |
| QM9 | r2 | MOLECULAR_FORMULA + INCHI + SELFIES + CANONICAL_SMILES |
| QM9 | u0 | SELFIES + MOLECULAR_FORMULA + INCHI + IUPAC_NAME + CANONICAL_SMILES |
| QM9 | u298 | SELFIES + MOLECULAR_FORMULA + INCHI + IUPAC_NAME + CANONICAL_SMILES |
| QM9 | zpve | SELFIES + MOLECULAR_FORMULA + INCHI + CANONICAL_SMILES |
| QM8 | E1-CAM | CANONICAL_SMILES |
| QM8 | E1-CC2 | SELFIES + MOLECULAR_FORMULA + INCHI + CANONICAL_SMILES |
| QM8 | E1-PBE0 | CANONICAL_SMILES |
| QM8 | E2-CAM | CANONICAL_SMILES |
| QM8 | E2-CC2 | CANONICAL_SMILES |
| QM8 | E2-PBE0 | CANONICAL_SMILES |
| QM8 | f1-CAM | CANONICAL_SMILES |
| QM8 | f1-CC2 | SELFIES + CANONICAL_SMILES |
| QM8 | f1-PBE0 | MOLECULAR_FORMULA + CANONICAL_SMILES + IUPAC_NAME |
| QM8 | f2-CAM | CANONICAL_SMILES |
| QM8 | f2-CC2 | CANONICAL_SMILES + IUPAC_NAME + MOLECULAR_FORMULA + SELFIES |
| QM8 | f2-PBE0 | CANONICAL_SMILES |
| ESOL | log solubility | IUPAC_NAME + MOLECULAR_FORMULA + SELFIES + INCHI + CANONICAL_SMILES |
| FreeSolv | expt | SELFIES + CANONICAL_SMILES + MOLECULAR_FORMULA + INCHI |
| Lipophilicity | y | SELFIES + MOLECULAR_FORMULA + INCHI + IUPAC_NAME + CANONICAL_SMILES |

Table 14: Top-2 molecular combinations for the MoleculeNet benchmark datasets.

| Dataset | Task | Molecular combination |
|---|---|---|
| BBBP | p_np | SELFIES + CANONICAL_SMILES |
| ClinTox | all | CANONICAL_SMILES + IUPAC_NAME + SELFIES + MOLECULAR_FORMULA |
| HIV | HIV_active | IUPAC_NAME + MOLECULAR_FORMULA + INCHI + CANONICAL_SMILES |
| BACE | Class | CANONICAL_SMILES |
| SIDER | all | IUPAC_NAME + MOLECULAR_FORMULA + INCHI + CANONICAL_SMILES + SELFIES |
| Tox21 | all | INCHI + CANONICAL_SMILES + SELFIES + IUPAC_NAME + MOLECULAR_FORMULA |
| QM9 | alpha | MOLECULAR_FORMULA |
| QM9 | cv | IUPAC_NAME + MOLECULAR_FORMULA + INCHI + CANONICAL_SMILES + SELFIES |
| QM9 | g298 | SELFIES + MOLECULAR_FORMULA + INCHI + IUPAC_NAME + CANONICAL_SMILES |
| QM9 | gap | CANONICAL_SMILES + IUPAC_NAME + MOLECULAR_FORMULA + SELFIES |
| QM9 | h298 | CANONICAL_SMILES + MOLECULAR_FORMULA + IUPAC_NAME + INCHI |
| QM9 | homo | SELFIES + CANONICAL_SMILES + INCHI + MOLECULAR_FORMULA |
| QM9 | lumo | INCHI + CANONICAL_SMILES + SELFIES + IUPAC_NAME + MOLECULAR_FORMULA |
| QM9 | mu | IUPAC_NAME + SELFIES + CANONICAL_SMILES + MOLECULAR_FORMULA |
| QM9 | r2 | SELFIES + INCHI + CANONICAL_SMILES + MOLECULAR_FORMULA + IUPAC_NAME |
| QM9 | u0 | CANONICAL_SMILES + MOLECULAR_FORMULA + IUPAC_NAME + INCHI |
| QM9 | u298 | CANONICAL_SMILES + MOLECULAR_FORMULA + IUPAC_NAME + INCHI |
| QM9 | zpve | CANONICAL_SMILES + MOLECULAR_FORMULA + IUPAC_NAME + INCHI |
| QM8 | E1-CAM | CANONICAL_SMILES + IUPAC_NAME + MOLECULAR_FORMULA + SELFIES |
| QM8 | E1-CC2 | INCHI + MOLECULAR_FORMULA + IUPAC_NAME + SELFIES + CANONICAL_SMILES |
| QM8 | E1-PBE0 | MOLECULAR_FORMULA + SELFIES + IUPAC_NAME + CANONICAL_SMILES |
| QM8 | E2-CAM | MOLECULAR_FORMULA + SELFIES + IUPAC_NAME + CANONICAL_SMILES |
| QM8 | E2-CC2 | MOLECULAR_FORMULA + SELFIES + IUPAC_NAME + CANONICAL_SMILES |
| QM8 | E2-PBE0 | MOLECULAR_FORMULA + SELFIES + IUPAC_NAME + CANONICAL_SMILES |
| QM8 | f1-CAM | IUPAC_NAME + CANONICAL_SMILES |
| QM8 | f1-CC2 | CANONICAL_SMILES |
| QM8 | f1-PBE0 | SELFIES + CANONICAL_SMILES + MOLECULAR_FORMULA + IUPAC_NAME |
| QM8 | f2-CAM | CANONICAL_SMILES + IUPAC_NAME + MOLECULAR_FORMULA + SELFIES |
| QM8 | f2-CC2 | CANONICAL_SMILES |
| QM8 | f2-PBE0 | CANONICAL_SMILES + IUPAC_NAME + SELFIES + MOLECULAR_FORMULA |
| ESOL | log solubility | INCHI + MOLECULAR_FORMULA + SELFIES + CANONICAL_SMILES + IUPAC_NAME |
| FreeSolv | expt | SELFIES + MOLECULAR_FORMULA + INCHI + IUPAC_NAME + CANONICAL_SMILES |
| Lipophilicity | y | MOLECULAR_FORMULA + IUPAC_NAME + INCHI + CANONICAL_SMILES + SELFIES |

the formulation string, respectively. In addition, each molecular notation was included with their

Table 15: Top-3 molecular combinations for the MoleculeNet benchmark datasets.

| Dataset | Task | Molecular combination |
|---|---|---|
| BBBP | p_np | MOLECULAR_FORMULA + IUPAC_NAME + CANONICAL_SMILES + INCHI + SELFIES |
| ClinTox | all | IUPAC_NAME + SELFIES + MOLECULAR_FORMULA |
| HIV | HIV_active | MOLECULAR_FORMULA + IUPAC_NAME + CANONICAL_SMILES + INCHI + SELFIES |
| BACE | Class | INCHI + MOLECULAR_FORMULA + IUPAC_NAME + SELFIES + CANONICAL_SMILES |
| SIDER | all | SELFIES + MOLECULAR_FORMULA + IUPAC_NAME + CANONICAL_SMILES |
| Tox21 | all | IUPAC_NAME + MOLECULAR_FORMULA + INCHI + CANONICAL_SMILES |
| QM9 | alpha | SELFIES + MOLECULAR_FORMULA + INCHI + IUPAC_NAME + CANONICAL_SMILES |
| QM9 | cv | CANONICAL_SMILES + MOLECULAR_FORMULA + INCHI + IUPAC_NAME + SELFIES |
| QM9 | g298 | CANONICAL_SMILES + MOLECULAR_FORMULA + IUPAC_NAME + INCHI |
| QM9 | gap | INCHI + MOLECULAR_FORMULA + IUPAC_NAME + SELFIES + CANONICAL_SMILES |
| QM9 | h298 | INCHI + MOLECULAR_FORMULA + IUPAC_NAME + CANONICAL_SMILES + SELFIES |
| QM9 | homo | CANONICAL_SMILES + MOLECULAR_FORMULA + IUPAC_NAME + INCHI + SELFIES |
| QM9 | lumo | INCHI + MOLECULAR_FORMULA + IUPAC_NAME + SELFIES + CANONICAL_SMILES |
| QM9 | mu | SELFIES + MOLECULAR_FORMULA + INCHI + IUPAC_NAME + CANONICAL_SMILES |
| QM9 | r2 | INCHI + MOLECULAR_FORMULA + IUPAC_NAME + SELFIES + CANONICAL_SMILES |
| QM9 | u0 | INCHI + MOLECULAR_FORMULA + IUPAC_NAME + CANONICAL_SMILES + SELFIES |
| QM9 | u298 | INCHI + MOLECULAR_FORMULA + IUPAC_NAME + CANONICAL_SMILES + SELFIES |
| QM9 | zpve | IUPAC_NAME + MOLECULAR_FORMULA + INCHI + CANONICAL_SMILES + SELFIES |
| QM8 | E1-CAM | SELFIES + MOLECULAR_FORMULA + INCHI + CANONICAL_SMILES |
| QM8 | E1-CC2 | INCHI + CANONICAL_SMILES + SELFIES + IUPAC_NAME + MOLECULAR_FORMULA |
| QM8 | E1-PBE0 | CANONICAL_SMILES + IUPAC_NAME + MOLECULAR_FORMULA + SELFIES |
| QM8 | E2-CAM | MOLECULAR_FORMULA + IUPAC_NAME + CANONICAL_SMILES |
| QM8 | E2-CC2 | MOLECULAR_FORMULA + IUPAC_NAME + CANONICAL_SMILES |
| QM8 | E2-PBE0 | INCHI + MOLECULAR_FORMULA + IUPAC_NAME + SELFIES + CANONICAL_SMILES |
| QM8 | f1-CAM | CANONICAL_SMILES + INCHI + SELFIES + IUPAC_NAME + MOLECULAR_FORMULA |
| QM8 | f1-CC2 | CANONICAL_SMILES + IUPAC_NAME + MOLECULAR_FORMULA + SELFIES |
| QM8 | f1-PBE0 | INCHI + MOLECULAR_FORMULA + IUPAC_NAME + SELFIES + CANONICAL_SMILES |
| QM8 | f2-CAM | MOLECULAR_FORMULA + SELFIES + IUPAC_NAME + CANONICAL_SMILES |
| QM8 | f2-CC2 | MOLECULAR_FORMULA + IUPAC_NAME + CANONICAL_SMILES |
| QM8 | f2-PBE0 | SELFIES + CANONICAL_SMILES + MOLECULAR_FORMULA + IUPAC_NAME |
| ESOL | log solubility | SELFIES + MOLECULAR_FORMULA + INCHI + IUPAC_NAME + CANONICAL_SMILES |
| FreeSolv | expt | MOLECULAR_FORMULA + IUPAC_NAME + CANONICAL_SMILES + INCHI + SELFIES |
| Lipophilicity | y | IUPAC_NAME + SELFIES + CANONICAL_SMILES + INCHI + MOLECULAR_FORMULA |

Table 16: Polymer membranes prediction performance. $R^2$ is used as evaluation metric, therefore, in this case higher values is better. **Blue** and **Orange** indicates best and second-best performing model, respectively.

| Method | Dataset | | |
|---|---|---|---|
| | $T_{d\frac{1}{2}}$ | $T_g$ | $\log(P_{CO_2})$ |
| Lasso Giro et al. (2023) | 0.81 | 0.90 | 0.87 |
| ElasticNet Giro et al. (2023) | 0.81 | 0.88 | 0.89 |
| Ridge Giro et al. (2023) | 0.82 | 0.90 | 0.90 |
| SPG-TED$_{289M}$ Soares et al. | 0.96 | 0.86 | 0.88 |
| STR-Bamba$_{426M}$ (Pre-trained) | 0.98±0.001 | 0.86±0.007 | 0.91±0.005 |
| STR-Bamba$_{426M}$ (Fine-tuned) | 0.98±0.003 | 0.91±0.008 | 0.96±0.001 |

Table 17: Gas permeability of polymers (CalTech) prediction. $R^2$ is used as evaluation metric, therefore, in this case higher values is better. **Blue** and **Orange** indicates best and second-best performing model, respectively.

| Method | Dataset | | | | | |
|---|---|---|---|---|---|---|
| | He | $H_2$ | $O_2$ | $N_2$ | $CO_2$ | $CH_4$ |
| RF (descriptors) Yang et al. (2022) | 0.73 | 0.74 | 0.75 | 0.74 | 0.38 | 0.75 |
| DNN ensemble(descriptors) Yang et al. (2022) | 0.87 | 0.88 | 0.89 | 0.90 | 0.90 | 0.89 |
| DNN ensemble(MFFs) Yang et al. (2022) | 0.91 | 0.90 | 0.92 | 0.91 | 0.90 | 0.88 |
| SPG-TED$_{289M}$ Soares et al. | 0.92 | 0.87 | 0.89 | 0.91 | 0.91 | 0.85 |
| STR-Bamba$_{426M}$ (Pre-trained) | 0.7±0.02 | 0.73±0.03 | 0.75±0.05 | 0.76±0.02 | 0.78±0.03 | 0.78±0.02 |
| STR-Bamba$_{426M}$ (Fine-tuned) | 0.92±0.01 | 0.91±0.01 | 0.91±0.004 | 0.91±0.004 | 0.90±0.01 | 0.90±0.01 |

respective special token. Finally, formulation compositions were also added after each molecule separated with the *<sep>* special token.

Table 18: Polymer ionic conductivity. MAE is used as evaluation metric, therefore, in this case lower is better. **Blue** and **Orange** indicates best and second-best performing model, respectively.

| Method | Dataset
Polymer ionic
conductivity |
|---|---|
| XGBoost Hatakeyama-Sato et al. (2023) | 1.09 |
| Chemprop Hatakeyama-Sato et al. (2023) | 1.08 |
| ChemArr Hatakeyama-Sato et al. (2023) | 1.00 |
| SPG-TED$_{289M}$ Soares et al. | **0.89** |
| STR-Bamba$_{426M}$ (Pre-trained) | 0.92±0.001 |
| STR-Bamba$_{426M}$ (Fine-tuned) | **0.89±0.004** |

Table 19: Gas permeability of polymers (NETL) prediction. MAE is used as evaluation metric, therefore, in this case lower is better. **Blue** and **Orange** indicates best and second-best performing model, respectively.

| Method | Dataset | | | | |
|---|---|---|---|---|---|
| | $CO_2$ | $CO_2/CH_4$ | $CH_4$ | $CO_2/N_2$ | $N_2$ |
| SOTA Tiwari et al. (2024) | 0.29 | 5.34 | 0.37 | 4.14 | 0.38 |
| SPG-TED$_{289M}$ Soares et al. | 0.29 | **4.71** | 0.35 | **3.89** | 0.31 |
| STR-Bamba$_{426M}$ (Pre-trained) | **0.25±0.01** | 6.22±0.13 | **0.34±0.01** | 4.05±0.09 | **0.28±0.01** |
| STR-Bamba$_{426M}$ (Fine-tuned) | **0.24±0.01** | **4.69±0.16** | **0.28±0.002** | **3.83±0.08** | **0.25±0.01** |

Table 20: Polymer refractive index prediction. RMSE is used as evaluation metric, therefore, in this case lower is better. **Blue** and **Orange** indicates best and second-best performing model, respectively.

| Method | Dataset
Refractive
index (n) |
|---|---|
| GPT-4 Hatakeyama-Sato et al. (2023) | 0.0310 |
| Boruta Hatakeyama-Sato et al. (2023) | 0.0339 |
| SPG-TED$_{289M}$ Soares et al. | **0.0210** |
| STR-Bamba$_{426M}$ (Pre-trained) | 0.0276±0.003 |
| STR-Bamba$_{426M}$ (Fine-tuned) | **0.0234±0.0057** |

Table 21: Polymer multi-task prediction. RMSE is used as evaluation metric, therefore, in this case lower is better. **Blue** and **Orange** indicates best and second-best performing model, respectively.

| Method | Dataset | | | | | | |
|---|---|---|---|---|---|---|---|
| | Polymer
Chain
Bandgap (Egc) | Polymer
Electron
Affinity (Eea) | Polymer
Bulk
Bandgap (Egb) | Polymer
Ionization
Energy (Ei) | Polymer
Dielectric
Constant (EPS) | Polymer
Refractive
Index (Nc) | Polymer
Crystallization
Tendency (Xc) |
| SOTA Kuenneth & Ramprasad (2023) | **0.44** | **0.28** | 0.49 | **0.39** | 0.52 | **0.09** | **16.57** |
| Xu et al. (2023) | 0.49 | 0.29 | | | | | |
| SPG-TED$_{289M}$ Soares et al. | | | **0.32** | **0.37** | **0.38** | **0.12** | **17.82** |
| STR-Bamba$_{426M}$ (Pre-trained) | 0.55±0.01 | 0.36±0.01 | 0.63±0.02 | 0.63±0.02 | 0.77±0.03 | 0.15±0.01 | 22.26±1.02 |
| STR-Bamba$_{426M}$ (Fine-tuned) | **0.43±0.01** | **0.18±0.01** | **0.40±0.02** | 0.44±0.02 | **0.48±0.02** | 0.13±0.01 | 18.98±0.74 |

Table 22: Copolymer electron affinity and ionization potential. RMSE is used as evaluation metric, therefore, in this case lower is better. **Blue** and **Orange** indicates best and second-best performing model, respectively.

| Method | Dataset | |
|---|---|---|
| | EA (eV) | IP (eV) |
| Neural Networks (Monomer) Aldeghi & Coley (2022) | 0.22 | 0.19 |
| Neural Networks (Polymer) Aldeghi & Coley (2022) | 0.18 | **0.16** |
| wD-MPNN Aldeghi & Coley (2022) | **0.03** | **0.03** |
| SPG-TED$_{289M}$ Soares et al. | **0.15** | **0.16** |
| STR-Bamba$_{426M}$ (Pre-trained) | 0.21±0.001 | 0.20±0.001 |
| STR-Bamba$_{426M}$ (Fine-tuned) | 0.18±0.001 | 0.17±0.001 |

Table 23: Glass-transition temperature prediction. MAE is used as evaluation metric, therefore, in this case lower is better. **Blue** and **Orange** indicates best and second-best performing model, respectively.

| Method | Dataset $T_g$ (K) |
|---|---|
| SOTA Long et al. (2024) | 53.02 (24.42) |
| SPG-TED$_{289M}$ Soares et al. | 9.56 |
| STR-Bamba$_{426M}$ (Pre-trained) | 4.85±0.04 |
| STR-Bamba$_{426M}$ (Fine-tuned) | 3.36±0.52 |

Table 24: Individual molecular representations performance for the electrolyte formulation tasks. RMSE is used as evaluation metric, therefore, in this case lower is better.

| Representation | Method | Dataset | |
|---|---|---|---|
| | | Li/Cu Half-Cell (LCE) | Li-I Full-Cell Battery (Capacity) |
| Molecular Formula | Pre-trained | 0.255±0.021 | 38.272±4.775 |
| Canonical SMILES | Pre-trained | 0.288±0.007 | 33.655±3.368 |
| IUPAC Name | Pre-trained | 0.235±0.017 | 33.174±3.007 |
| InChI | Pre-trained | 0.295±0.028 | 40.310±1.899 |
| SELFIES | Pre-trained | 0.248±0.023 | 38.557±0.933 |
| Molecular Formula | Fine-tuned | 0.247±0.019 | 40.719±6.970 |
| Canonical SMILES | Fine-tuned | 0.231±0.033 | 35.022±9.222 |
| IUPAC Name | Fine-tuned | 0.201±0.011 | 45.730±6.182 |
| InChI | Fine-tuned | 0.236±0.034 | 32.496±7.066 |
| SELFIES | Fine-tuned | 0.214±0.031 | 42.389±5.105 |

