# OpenReview forum: "STR-Bamba: Multimodal Molecular Textual Representation Encoder-Decoder Foundation Model"
_ICLR.cc/2026/Conference — Submitted to ICLR 2026_

### Official Review · Reviewer_ugbg · 2025-10-27

**Soundness:** 2
**Presentation:** 3
**Contribution:** 2
**Rating:** 4
**Confidence:** 4

**Summary:**

The authors present STR‑Bamba, a 426M‑parameter encoder–decoder chemical foundation model that combines Transformer and Mamba‑2 layers in a hybrid architecture, trained on 588M samples spanning seven textual molecular modalities (SMILES, SELFIES, formula, IUPAC, InChI, serialized polymer graph, electrolyte formulations). A unified tokenizer with modality tags is used. Pre‑training proceeds in two phases: encoder pretraining (masked token + sentence-level equivalence to align representations across modalities), then decoder training conditioned on encoder embeddings. The paper reports analyses of the latent space (t‑SNE + K‑means across modalities) and downstream property prediction across 29 datasets as well as translation between representations.

**Strengths:**

* Comprehensive modality coverage in a single vocabulary with a practical tokenizer.
* Large‑scale pretraining and efficient architecture likely to benefit long‑context chemistry tasks.
* Latent alignment evidence: 7‑cluster structure matches the number of modalities; quantitative clustering scores reported. (Fig. 2 & Table 2 in the results section)

**Weaknesses:**

* Limited baselines/ablations: Need apples‑to‑apples comparisons vs. similarly sized Transformer‑only and Mamba‑only models under controlled token budgets. Architectural choices should be empirically justified.
* Attribution of gains: Separate the effects of multimodality from scale and architecture via controlled experiments.
* Task breadth: Polymer/electrolyte tasks are highlighted in pretraining; ensure strong downstream validations for those domains, not only small molecules.

**Questions:**

* How does STR‑Bamba perform vs. a Transformer‑only encoder–decoder with the same parameter count and token budget? Can you provide a controlled ablation?
* What is the contribution of each modality? Can you report leave‑one‑modality‑out ablations on downstream tasks?
* For translation tasks, can you report exact‑match and structure‑match rates with CIs across held‑out molecules, and failure analyses (e.g., stereochemistry)?
* How are the representations aligned? If I understand correctly, Figure 2(a) shows that each modality is clustered separately than other modalities, even for the sam molecule. What exactly do you mean by "alignment of modalities"?

Comment:
* Line 224: "We used 396 and 8 NVIDIA A100 (40GB) GPUs to train
phase 1 and phase 2, respectively.": What does 'We used 396 and 8 NVIDIA A100' mean? A typo?

---

### Official Review · Reviewer_vsgh · 2025-10-29

**Soundness:** 3
**Presentation:** 3
**Contribution:** 2
**Rating:** 4
**Confidence:** 4

**Summary:**

The authors propose to integrate the multiple molecular notations into one pretrained model, to complement the important molecular information across different molecular notations. Based on this, the authors pretrain a hybrid model that utilizing both Transformer and Mamba architectures. In experiments part, STR-Bamba generally outperform the existing baselines under various tasks.

**Strengths:**

1. The presentation is clear and the paper is easy to follow.

2. The experiments across 29 benchmark datasets are comprehensive to validate the effectiveness of the proposed method.

3. The motivation is clear to combine all the molecular notations to complement all molecular information.

**Weaknesses:**

1. Although the motivation sounds reasonable, can authors provide a specific example to demonstrate the importance of incorporating each kind of molecular notation? It will be of great help to demonstrate which notation includes which kinds of information. And I am also wondering if it is necessary to incorporate all kinds of notation? Is it possible the information from one notation has been supplemented by the other notations?

2. The authors claim the proposed method is inference-effienct. However, it seems no such experiment has been conducted to support such claim.

3. The ablation study or analysis of introducing the multiple molecular notations is missing. While the authors have conducted some experiments (latent space and notation translation), the necessity of introducing the comprehensive molecular notations is still not convincing.

4. The technical novelty is relatively weak. The model is basically adapted from existing works.

**Questions:**

See Weaknesses.

---

### Official Review · Reviewer_WHBW · 2025-11-01

**Soundness:** 3
**Presentation:** 3
**Contribution:** 3
**Rating:** 6
**Confidence:** 4

**Summary:**

This paper proposes STR-Bamba426M, a hybrid encoder-decoder model for molecular language modeling that combines Transformer layers with Mamba-2 state-space layers. The model is designed to handle multiple molecular textual representations (SMILES, SELFIES, SPG, molecular formula, InChI, IUPAC names, and electrolyte formulations) within a unified vocabulary. The authors implement a custom tokenizer with modality-specific preprocessing and train the model on a large dataset comprising 117M small molecules, 2M polymer structures, and 258 electrolyte formulations. The model leverages shared token and sentence embeddings to align different representations of the same molecule, and employs cross-attention layers in the decoder to generate valid molecular sequences. Pre-training is performed in two stages: first, the encoder learns embeddings and aligns molecular formats; second, the decoder is trained to reconstruct molecular sequences conditioned on the encoder embeddings.

**Strengths:**

* Multimodal molecular representation: The model can handle multiple types of molecular textual inputs simultaneously, which may allow richer information capture compared to single-representation models.

* Shared embeddings and alignment strategy: The use of token + sentence embeddings and alignment across molecular formats is a thoughtful design for improving the encoder’s representation quality.

* Large-scale dataset: The dataset is extensive, including both small molecules and polymers, which can help in testing generalization across chemical domains.

**Weaknesses:**

* Questionable need for long-range modeling: The justification for using Mamba-2 state-space layers to model “longer context lengths” is weak, because most molecular sequences (even when concatenating multiple representations) are relatively short compared to genomic sequences or long text, which state-space models were originally designed for. The added complexity may not be necessary and could likely be replaced with a simpler Transformer-only architecture.

* Data quality issues: Synthetic polymer structures may include non-viable molecules; the impact of this on model performance is not well analyzed.

* Tokenizer and vocabulary generalization: It is unclear how well the custom tokenizer handles rare molecules or new unseen formats. There is also no discussion on potential overfitting to frequent substructures.

**Questions:**

* The authors mention that InChI tokens are tokenized by extracting atom-associated numbers. Given that InChI tokens are highly structured and interdependent, could the authors clarify how this tokenization preserves the chemical and structural semantics inherent in InChI?

* What is the inference speed and memory requirement for STR-Bamba426M compared to simpler Transformer models for typical molecular inputs?

---

### Official Review · Reviewer_VYHY · 2025-11-04

**Soundness:** 3
**Presentation:** 2
**Contribution:** 3
**Rating:** 4
**Confidence:** 4

**Summary:**

This paper introduces STR-Bamba426M, a 426-million-parameter multimodal encoder–decoder foundation model for molecular data. Unlike prior chemical LLMs that rely on a single representation (e.g., SMILES or SELFIES), STR-Bamba unifies seven textual modalities particularly SMILES, SELFIES, IUPAC, InChI, molecular formula, serialized polymer graphs (SPG), and electrolyte formulations, within a shared vocabulary and hybrid Transformer + Mamba-2 architecture. The model is pre-trained on 588 million molecular samples (≈29 billion tokens), using PubChem, synthetic polymers, and electrolyte data, then evaluated on MoleculeNet, polymer property prediction, and electrolyte formulation benchmarks. The authors claim near- or better-than-SOTA results on 9 of 11 MoleculeNet tasks, strong performance across 26 polymer tasks, and effective representation translation among molecular notations.

**Strengths:**

There is a comprehensive pre-training corpus (588 M samples; 29 B tokens) across small molecules, polymers, and electrolytes.

The hybrid architecture combines both long-context efficiency (Mamba-2) and attention flexibility.

There is a unified tokenizer for multiple modalities, which is a significant engineering contribution.

Experiment results are thorough with extensive benchmarking across small molecules, polymers, and electrolytes.

There is demonstrated cross-representation latent alignment like t-SNE + K-means.

**Weaknesses:**

This work lacks empirical rigor. Specifically, there is no ablation isolating modality impact or architecture components.

Moreover, the comparisons are lacking evaluation against state of the art models. Specifically, head-to-head with MolX (KDD 2024), MolXPT, or Uni-Mol 3D under identical tuning comparisons are omitted.

This work is also not reproducible. Specifically, it is missing training FLOPs, parameter scaling laws, or energy footprint. Code/weights unavailable, tokenization details are also incomplete.

The writing can also be improved. It is too long, verbose, and sometimes repetitive for a 9-page main paper format.

Some critical works are also missing for citation. These include the below:

- Le, Khiem, Zhichun Guo, Kaiwen Dong, Xiaobao Huang, Bozhao Nan, Roshni Iyer, Xiangliang Zhang, Olaf Wiest, Wei Wang, Ting Hua & Nitesh V. Chawla. MolX: Enhancing Large Language Models for Molecular Understanding With A Multi-Modal Extension. Proceedings of the 2025 ACM SIGKDD International Conference on Knowledge Discovery & Data Mining (MLoG-GenAI@KDD ’25), ACM, 2025.

- Liu, Yong, et al. MolX: Enhancing Large Language Models for Molecular Understanding with a Multi-Modal Extension. KDD Proceedings, 2024.

- Soares, Eduardo A., et al. “An Open-Source Family of Large Encoder–Decoder Foundation Models for Chemistry.” Communications Chemistry, vol. 8, no. 1, 2025.

**Questions:**

How do you prevent data leakage between PubChem-derived pre-training and MoleculeNet test sets?

What is the relative contribution of Mamba-2 layers vs. Transformers? Please include ablations.

How sensitive are results to tokenizer vocabulary size (5 k) and modality balance?

Can the model generalize to unseen representations (e.g., CXSMILES or graphical forms)?

What is the computational cost (GPU-hours, energy) of pre-training vs. SSM-only baselines?

How does translation fidelity compare with Chemformer or MolXPT using BLEU or Tanimoto metrics?

Will you release the code and tokenizer to the community?

**Details Of Ethics Concerns:**

This work uses open PubChem and literature data but no explicit license verification.

The environment cost of running this model should also be taken into consideration. There are 400+ GPUs over two phases (396 A100s + 8 A100s) suggests > 1 MWh consumption; no disclosure.

---

### Meta-Review · Area_Chair_ropP · 2026-01-10

**Summary:**

The reviewers collectively point out a significant lack of ablation experiments to justify the chosen architecture. Critical concerns include missing baseline comparisons, limited technical novelty, and marginal performance improvements.

**Reviewer Concerns:**

Since the authors failed to provide a rebuttal, all reviewer concerns remain outstanding, eg. missing baseline comparisons, limited technical novelty, and marginal performance improvements.

**Reviewer Scores:**

The scores (4, 6, 4, 4) remain unchanged or even decreased as the authors did not respond to any of the specific critiques regarding methodology, baselines, or reproducibility.

---

### Decision · Program_Chairs · 2026-01-26

Reject